

# Seasonal extrema of sea surface temperature in CMIP6 models

Yanxin Wang[1], Karen J. Heywood[1], David P. Stevens[1], and Gillian M. Damerell[1]

[1]Centre for Ocean and Atmospheric Sciences, University of East Anglia, Norwich, UK

**Correspondence:** Yanxin Wang (Yanxin.Wang@uea.ac.uk)

**Abstract.** CMIP6 model sea surface temperature (SST) seasonal extrema averaged over 1981-2010 are assessed against the World Ocean Atlas (WOA18) observational climatology. We propose a mask to identify and exclude regions of large differences between three commonly-used climatologies. The biases in SST seasonal extrema are largely consistent with the annual mean SST biases. However, the amplitude and spatial pattern of SST bias vary seasonally in the 20 CMIP6 models assessed. Large seasonal variations in the SST bias occur in eastern boundary upwelling regions, polar regions, the North Pacific and eastern equatorial Atlantic. These results demonstrate the importance of evaluating model performance not simply against annual mean properties. Models with greater vertical resolution in their ocean component typically demonstrate better representation of SST extrema, particularly seasonal maximum SST. No significant relationship with horizontal ocean model resolution is found.

## 1 Introduction

Seasonal extrema of sea surface temperature (SST) are important for the global climate system. SST seasonal maxima influence the formation and intensity of tropical cyclones (Palmen, 1948; Dare and McBride, 2011; Holland, 1997; Sun et al., 2017) and may be associated with marine heatwaves, which can cause damage to marine ecosystems worldwide, including biomass decrease, bleaching of coral reefs, and deaths of marine animals (Cheung and Frölicher, 2020; Hughes et al., 2018; Jones et al., 2018). SST seasonal minima are closely linked to formation of sea ice and determine the properties of intermediate and deep water. Heat loss in winter allows surface water to subduct into the deep ocean, important for thermohaline circulation. Therefore, future projections of tropical cyclones, heatwaves, water mass formation or sea ice extent require our models to have a realistic representation of SST seasonal extrema.

Typically, however, evaluations of climate model historical runs focus on annual or long-term mean SST, revealing common biases across many models (Wang et al., 2014; Flato et al., 2013). Assessments of model performance in simulating SST seasonal cycles are less common, and are often only regional. For example, a marked seasonal variability of SST warm bias in the eastern tropical Atlantic has been documented in CMIP5 (Coupled Model Intercomparison Project Phase 5) and CMIP6 (CMIP Phase 6) models (Prodhomme et al., 2019; Richter et al., 2014; Richter and Tokinaga, 2020). In these models, the eastern tropical Atlantic warm bias is maximum in boreal summer (June-July-August), which has been attributed to the largest wind biases occurring during spring (Richter et al., 2012; Richter and Tokinaga, 2020). Similarly, CMIP6 model SST cold biases in the North Pacific subtropics vary seasonally (Zhu et al., 2020). Song and Zhang (2020) suggested that the CMIP5 multi-model mean has seasonally dependent SST biases in the northeastern Pacific Ocean, with a warm bias during summer





and a cold bias during winter, which they argued was caused by poorly simulated North American monsoon winds. Wang et al. (2014) showed that the amplitude of CMIP5 multi-model mean SST biases varies seasonally and therefore an accurate annual mean SST does not guarantee accurate seasonal extrema or seasonal cycle. Here we evaluate the seasonal cycle globally

in 20 state-of-the-art CMIP6 climate models, to provide a foundation for model SST bias identification and future reduction. By presenting maps of SST bias in seasonal extrema for each model, we highlight the care needed in selecting these models for future climate projections in particular regions. Section 2 introduces the models and the analysis techniques, including evaluation of uncertainties in global observational climatologies. Section 3 presents and discusses the biases in SST maxima and minima, and explores possible causes.

## 2   Data and Methods

The historical runs of 20 models (table 1) were averaged over 1981-2010 to create monthly mean climatologies for each model. The first ensemble member (r1i1p1f1) is used where available; we choose r1i1p1f3 for HadGEM3-GC3-LL and HadGEM3-GC3-MM; r1i1p1f2 for UKESM1-0-LL. The models include those incorporating biogeochemical cycling (earth system models) as well as conventional climate models. The ocean vertical coordinate is typically z-level (or the related $z^*$) but some

models use isopycnal, sigma or hybrid coordinates (table 1). The total number of levels and thickness of top grid cell are used as proxies for ocean vertical resolution. The global averaged thickness of top grid cell in INM-CM5-0 was calculated using the sigma coordinates and bottom topography obtained from E.M.Volodin (personal communication). The thickness of the top grid cell in other models was obtained from the references cited in table 1.

To examine the seasonal cycle of SST, most studies picked specific months to represent summer and winter (e.g., Zhang and

Zhao (2015); Liu et al. (2020)). However, model seasonal cycles may be out of phase with observations and observed maxima and minima occur in different months in different regions. Instead, here we take the maximum and minimum SST of the monthly mean climatologies ($T_{max}$ and $T_{min}$) at each grid point, identifying which months they occur in, for both model and observation. $T_{max}$ and $T_{min}$, plus the annual mean SST ($T_{mean}$) and the range of the seasonal cycle ($T_{cycle} = T_{max} - T_{min}$) from the model climatologies are compared with the World Ocean Atlas 2018 (WOA18) observational climatology on a grid

spacing of $0.25° \times 0.25°$ (Locarnini et al., 2018), which covers the period from 1981 to 2010. The model fields were interpolated to the same grid as WOA18. Biases are defined as model values minus WOA18 values. For the multi-model mean, at each grid point we average $T_{max}$, $T_{min}$, $T_{mean}$ and $T_{cycle}$ across the 20 CMIP6 models. To quantify the performance of CMIP6 models, we calculated the area-weighted root mean square error of the model against WOA18 (henceforth RMSE) for global SST.

Since there is some uncertainty in observational climatologies because of sparse sampling, instrumental error, quality control

or gridding techniques, we compared three recent climatologies: WOA18, WOCE-Argo Global Hydrographic Climatology (WAGHC)(Gouretski, 2018) (covering the time period 1985-2016), and HadISST (Rayner et al., 2003) (covering the time period 1981-2010). Any grid points where the maximum difference in $T_{max}$ or $T_{min}$ between the three climatologies is larger than 2°C are considered uncertain for that variable, and these grid points are excluded from our assessment. Any grid points which did not have values for all 12 months for at least two climatologies are also excluded. For $T_{mean}$ and $T_{cycle}$, we exclude





**Table 1.** The 20 CMIP6 models used in this study; the horizontal resolution of their ocean component; ocean vertical coordinate (z: traditional height coordinate; $z^*$: rescaled height coordinate for more accurate representation of free-surface variations; $\rho$: isopycnic coordinate; $\sigma$: terrain-following sigma coordinate; multiple symbols refer to a hybrid coordinate); total number of ocean vertical levels; thickness of the ocean top grid cell; and references.

| Model | Horizontal resolution | Vertical coordinate | Total levels | Top grid thickness | References |
|---|---|---|---|---|---|
| ACCESS-CM2 | 100 km | $z^*$ | 50 | 10 m | Bi et al. (2020) |
| ACCESS-ESM1-5 | 100 km | $z^*$ | 50 | 10 m | Law et al. (2017) |
| AWI-CM-1-1-MR | 25 km | z-$\sigma$ | 46 | 5 m | Semmler et al. (2020) |
| BCC-CSM2-MR | 50 km | z | 40 | 10 m | Wu et al. (2019) |
| BCC-ESM1 | 50 km | z | 40 | 10 m | Wu et al. (2020) |
| CESM2 | 100 km | z | 60 | 10 m | Danabasoglu et al. (2020) |
| CanESM5 | 100 km | z | 45 | 6 m | Swart et al. (2019) |
| E3SM-1-0 | 50 km | $z^*$ | 60 | 10 m | Golaz et al. (2019) |
| GFDL-CM4 | 25 km | $z^*$-$\rho$ | 75 | 2 m | Held et al. (2019) |
| GISS-E2-1-G | 100 km | z | 40 | 10 m | Kelley et al. (2020) |
| GISS-E2-1-H | 100 km | z-$\rho$-$\sigma$ | 32 | 10 m | Kelley et al. (2020) |
| HadGEM3-GC31-LL | 100 km | $z^*$ | 75 | 1 m | Andrews et al. (2020) |
| HadGEM3-GC31-MM | 25 km | $z^*$ | 75 | 1 m | Andrews et al. (2020) |
| INM-CM5-0 | 50 km | $\sigma$ | 40 | 7.3 m | Volodin et al. (2017) |
| IPSL-CM6A-LR | 100 km | $z^*$ | 75 | 2 m | Boucher et al. (2020) |
| MIROC6 | 100 km | z-$\sigma$ | 62 | 2 m | Tatebe et al. (2019) |
| MPI-ESM1-2-HR | 50 km | z | 40 | 12 m | Müller et al. (2018) |
| NorESM2-MM | 100 km | $\rho$ | 53 | 2.5 m | Seland et al. (2020) |
| SAM0-UNICON | 100 km | z | 60 | 10 m | Park et al. (2019) |
| UKESM1-0-LL | 100 km | $z^*$ | 75 | 1 m | Sellar et al. (2019) |

any points where either $T_{max}$ or $T_{min}$ is excluded. 4%, 3%, 4% and 4% of the ocean's surface area is excluded for $T_{max}$, $T_{min}$, $T_{mean}$ and $T_{cycle}$ respectively. Similarly, for the timing of $T_{max}$ and $T_{min}$, any grid points which did not have values for at least two climatologies or their maximum difference between climatologies in timing is larger than 2 months are excluded. In our global maps, these points are masked, and in calculations of global and regional metrics, these points are excluded.

## 3  Results and Discussion

### 3.1  Model representation of SST extrema

For the multi-model mean, $T_{max}$ and $T_{min}$ have larger global RMSEs than $T_{mean}$ (Fig. 1), as SST biases with opposite signs in different seasons compensate each other when calculating the annual mean. Similarly, the $T_{max}$ and $T_{min}$ global RMSEs of the multi-model mean are smaller than the RMSEs of individual models (Figs. 1b-c, 2, 3). Therefore, a small bias in $T_{mean}$



does not guarantee a realistic $T_{max}$ or $T_{min}$; a small bias in a multi-model mean does not guarantee good performance of
individual models.

The magnitudes of biases in $T_{max}$ and $T_{min}$ vary from model to model (Figs. 2, 3). The multi-model mean has RMSE
less than 1°C in both $T_{max}$ and $T_{min}$ (0.89°C and 0.87°C respectively). Most models have $T_{max}$ and $T_{min}$ RMSEs between
1°C and 2°C. Only HadGEM3-GC31-LL and GFDL-CM4 have $T_{max}$ RMSE less than 1°C (0.94°C and 0.93°C respectively).
GISS-E2-1-H has the largest $T_{max}$ RMSE of 1.89°C and MIROC6 has the largest $T_{min}$ RMSE of 1.62°C (Figs. 2, 3).

The bias in the timing of $T_{max}$ and $T_{min}$ is within one month in most of the global ocean in most models (Figs. 4, 5, 6).
In the multi-model mean, $T_{max}$ and $T_{min}$ occur one month earlier than in WOA18 for most of the global ocean, whereas in
some parts of the Arabian Sea and equatorial regions, they occur one month later (Fig. 4). It demonstrates that seasonal cycles
in CMIP6 models are out of phase with observations. In regions where monsoon prevails (e.g. the northwestern Indian Ocean),
the timing bias suggests a bias in the onset of summer monsoon.

$T_{max}$ and $T_{min}$ biases vary with latitude (Figs. 1b-c, 2, 3, 7g-h). Typically, the RMSE of $T_{max}$ at 30°-80° is 1-2°C larger
than at low latitudes (between 30°N and 30°S) (Fig. 7g). For GISS-E2-1-H, GISS-E2-1-G, BCC-CSM2-MR, BCC-ESM1 and
IPSL-CM6A-LR, $T_{max}$ RMSEs at 30°N-80°N are about 3°C larger than at low latitudes. A similar pattern is seen for $T_{min}$,
but the variation of biases with latitude is much smaller than for $T_{max}$ (Fig. 1c, 7h). Flato et al. (2013) found a similar result for
some CMIP5 models, with larger zonal mean biases in $T_{mean}$ between 30° and 70° than at other latitudes. The larger biases,
and greater difference between $T_{max}$ and $T_{min}$, at mid-high latitudes (greater than 30° in both hemispheres) may be explained
by the large seasonal cycle of mixed layer depth there. Shallower summer mixed layers have smaller heat capacity, thus a small
error in heat fluxes or mixing processes can result in a large bias for $T_{max}$, though this will be modulated by any seasonal
biases in mixed layer depth. The larger inter-model biases in $T_{max}$ than in $T_{min}$ can be explained by the shallower mixed layer
in summer, which can amplify SST biases due to biases in surface heat flux. The difference between biases in $T_{max}$ and $T_{min}$
leads to biases in $T_{cycle}$ (Fig. 1d). The RMSE of $T_{cycle}$ at low latitudes is typically 1°C, whereas at mid-high latitudes it is
larger, particularly in the Northern Hemisphere (Fig. 7i). The $T_{cycle}$ RMSE in IPSL-CM6A-LR and MIROC6 reaches 4°C at
high latitudes (Fig. 7i).

In polar regions, there are very small $T_{min}$ biases (Figs. 1c, 3, 7h) except for MIROC6 in the Antarctic. Winter SSTs are
close to freezing, but cannot go below freezing because sea ice forms instead. If models have realistic freezing points, $T_{min}$
biases will be small. Some models have salinity-dependent freezing points (Beaumet et al., 2019) in which case a salinity bias
could cause a bias in temperature. $T_{min}$ biases in the Arctic are larger than in the Antarctic (Figs. 1c, 7e-f), which suggests
larger saline biases in the Arctic.

In the subtropical North Pacific, the SST cold bias is typically 0.5-1°C smaller in $T_{max}$ than $T_{min}$, which leads to a too
large $T_{cycle}$ (Figs. 1b-d, 2, 3). Zhu et al. (2020) showed a similar seasonal SST cold bias in the CMIP6 multi-model mean, but
not in the CMIP5 multi-model mean. Underestimated surface shortwave radiation and too strong westerly winds in the CMIP6
multi-model mean (Lyu et al., 2020; Li et al., 2020) are possible reasons for the year round cold bias. The shortwave radiation
bias is likely related to the bias of low-level cloud in the subtropics (Burls et al., 2017; Li and Xie, 2012), and its associated
cold bias is smaller in winter when there is less solar radiation. The westerly winds cool the surface through latent heat flux





**Figure 1.** Biases (model minus climatology) of multi-model mean in (a) $T_{mean}$ (b) $T_{max}$ (c) $T_{min}$ (d) $T_{cycle}$. Black dots mark grid points excluded from our analysis, as described in section 2. The numbers indicate the global RMSE ($^{\circ}$C).

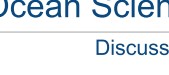
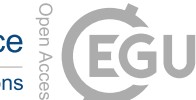

**Figure 2.** (a) $T_{max}$ in WOA18 and (b-u) $T_{max}$ model biases. Black dots mark grid points excluded from our analysis, as described in section 2. The numbers on (b-u) indicate the global RMSE of $T_{max}$. Red lines in (a) are 30°N and 30°S. Note that the range of bias color bar is twice as much as in Fig. 1.



**Figure 3.** As in Fig. 2, but for $T_{min}$.





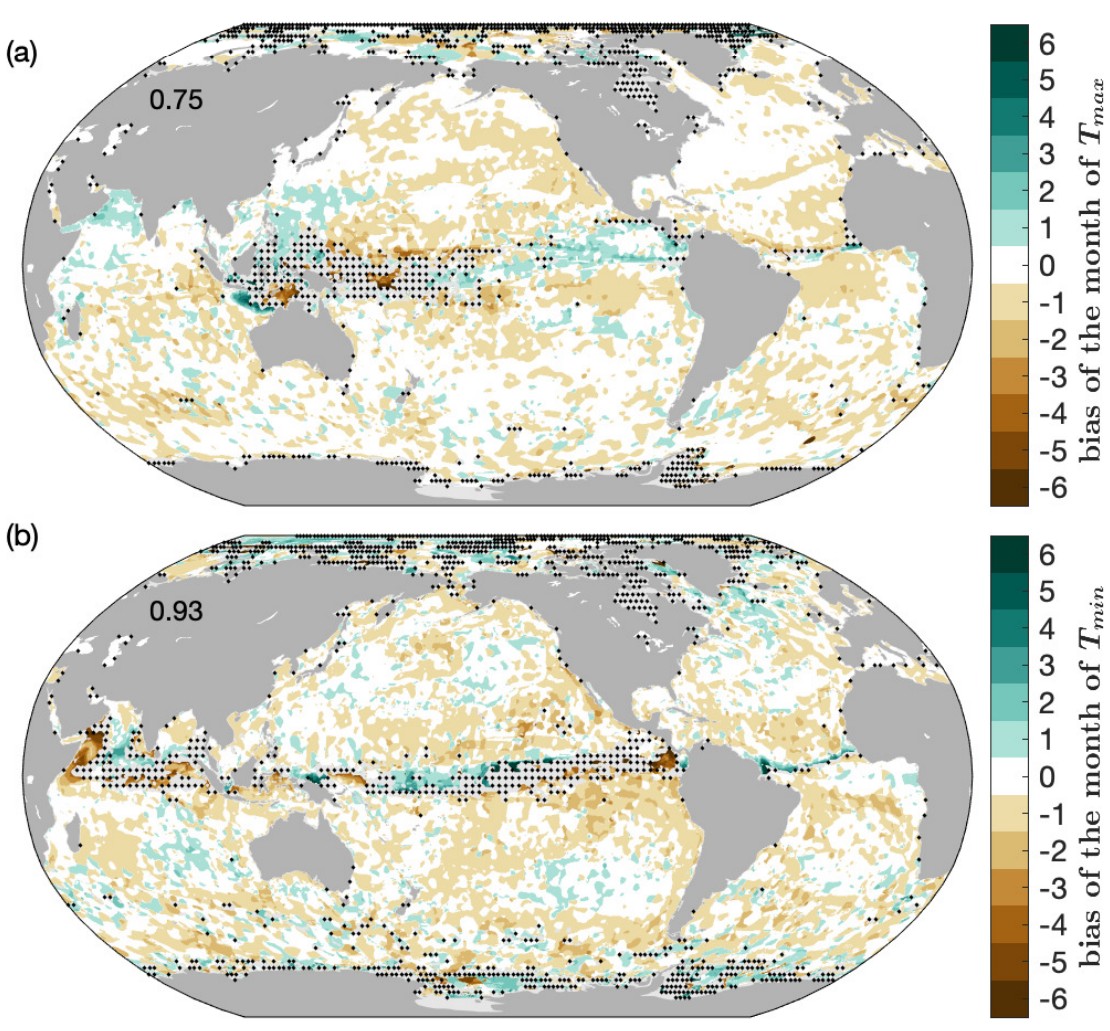

**Figure 4.** Biases in the timing of (a) $T_{max}$ and (b) $T_{min}$ in the multi-model mean. Black dots mark grid points excluded from our analysis, as described in section 2.





**Figure 5.** (a) Timing of $T_{max}$ in WOA18 and (b-u) biases in the timing of $T_{max}$ in models. Black dots mark grid points excluded from our analysis, as described in section 2.





**Figure 6.** As in Fig. 5, but for timing of $T_{min}$.





**Figure 7.** Monthly time series of area-weighted mean SST over (a) western equatorial Pacific (5°S - 5°N, 140°E - 160°W), (b) northwestern Indian Ocean (60 - 70°E, 10 - 20°N), (c) subtropical Southern Hemisphere (30° - 40°S), (d) subtropical Northern Hemisphere (30 - 40°N), (e) Arctic (70 - 80°N), (f) Antarctic (70 - 80°S), and area-weighted RMSE in 10° bands for (g) $T_{max}$, (h) $T_{min}$, (i) $T_{cycle}$. Y-axis range is same for (a-f).



and southward ocean advection due to Ekman transport. The latent heat loss is larger in summer (Yu, 2007), while the ocean
heat advection is larger in winter when meridional SST gradients are greater.

SST biases are seasonally dependent in the northeastern Pacific Inter Tropical Convergence Zone (ITCZ) (Figs. 1b-c, 2, 3).
For the multi-model mean, there is a warm bias in $T_{max}$ which exceeds 2°C and a cold bias in $T_{min}$ of 0.5-1.5°C. Similar
seasonal biases exist in CMIP5 models and were linked to an easterly wind bias throughout the year there (Song and Zhang,
2020). A coarse atmospheric model resolution smooths out the elevation difference between mountains and oceans, which
allows easterly trade winds to cross the mountains, leading to the easterly wind bias (Song and Zhang, 2020). An easterly bias
of annual mean wind was found in the CMIP6 multi-model mean (Li et al., 2020; Lyu et al., 2020). If the easterly bias exists
throughout the year, it can explain the seasonal SST bias we found. During winter-spring, the northeastern Pacific ITCZ is
dominated by easterly winds, so overly strong easterly winds enhance surface evaporation and lead to cold biases. In contrast,
during summer-autumn when westerly winds dominate, the simulated wind is too weak, which causes the warm bias. The
northeastern Pacific is a region where tropical cyclones and heatwaves occur (Gilford et al., 2017; Frölicher and Laufkötter,
2018), so a warm bias of over 2°C in $T_{max}$ may lead to overprediction of tropical cyclones and heatwaves.

The multi-model mean has a cold bias in $T_{max}$ and a warm bias in $T_{min}$ over the Northwest Pacific, leading to a too small
$T_{cycle}$ (bias of more than 2°C) (Figs. 1b-d). The warm bias in winter can be seen in many models, especially in ACCESS-
ESM1-5, BCC-ESM1, CanESM5 and INM-CM5-0 (Fig. 3). Models with a warm bias in $T_{min}$ are likely to generate overly
intense winter storms, as warm SSTs will increase the storm energy source. Greeves et al. (2007) demonstrated that there was
a clear link in the Hadley Centre models between winter SST warm bias to the east of Japan and increased storm intensity in
the region. The winter warm bias east of Japan was also found in a CMIP5 multi-model mean (Wang et al., 2018), but from
our results the warm bias extends further east (Fig. 1c).

The large cold biases at northern hemisphere high latitudes in BCC-CSM2-MR, BCC-ESM1, GISS-E2-1-G and GISS-E2-
1-H, are typically 2-5°C smaller in $T_{min}$ than in $T_{max}$ (Figs. 2, 3, 7g-h). These cold biases have been previously linked to
cloud biases. The negative cloud radiative forcing is excessive in BCC-CSM2-MR (Wu et al., 2019) and BCC-ESM1 (cloud
simulation likely to be similar to BCC-CSM2-MR), while overestimated low-cloud cover in GISS-E2-1-G and GISS-E2-1-H
(Kelley et al., 2020) blocks more of the incoming solar radiation. As solar radiation is negligible at high latitudes in winter, the
SST cold bias due to cloud bias is much smaller in winter than in summer, consistent with our results. Deep winter mixed layer
depths and SSTs close to freezing likely also contribute to the smaller cold biases in $T_{min}$ than in $T_{max}$ at high latitudes.

In most models there is a warm $T_{mean}$ bias in the Southern Ocean, commonly attributed to excessive short wave radiation
linked to underestimated cloud (Hyder et al., 2018). MIROC6 has an underestimated mid-level cloud cover (Tatebe et al.,
2019); GISS-E2-1-G and GISS-E2-1-H have an underestimated short wave cloud radiative forcing (Kelley et al., 2020), and
hence they have pronounced warm biases in the Southern Ocean (Figs. 2, 3). The warm bias is larger for $T_{max}$ than $T_{min}$ (Figs.
1b-c, 2, 3, 7g-h), because the lack of incoming solar radiation in winter means cloud biases have minimal effect on surface
solar insolation. Shallower mixed layer depths in summer will also tend to enhance any bias in incoming solar insolation. The
larger warm bias in $T_{max}$ than $T_{min}$ results in a sea ice extent that is too small in most CMIP6 models, especially in summer



(Beadling et al., 2020; Shu et al., 2020). As mode and intermediate waters primarily form within the winter mixed layer of the Antarctic Circumpolar Current (Talley, 1999), the $T_{min}$ warm bias can influence global ocean stratification.

MIROC6 stands out with the largest warm bias in the Southern Ocean (Figs. 2m, 3m), with a $T_{max}$ RMSE between 3 and 5°C and $T_{min}$ RMSE between 2 and 3°C at 50-80° S (Fig. 7g). The largest biases in MIROC6 occur in regions where there should be sea ice and where the deep ocean is ventilated. Beadling et al. (2020) found that MIROC6 has the lowest Southern Ocean sea ice extent among CMIP6 models in both summer and winter, and Tatebe et al. (2019) revealed annual warm biases exceeding 2°C in the intermediate and deep layers of MIROC6.

In eastern boundary upwelling regions (especially the Benguela and Humboldt Currents), most models have a seasonal warm bias that is 1-5°C smaller in $T_{max}$ than $T_{min}$ (Figs. 1b-c, 2, 3). Richter (2015) suggested that underestimation of stratocumulus cloud and insufficient upwelling due to overly weak winds contribute to the warm bias in eastern boundary upwelling regions. The warm bias we found therefore is likely associated with the underestimated surface shortwave radiation and overly weak upwelling-favourable winds in CMIP6 models identified by Li et al. (2020). The warm bias may lead to excessive precipitation

in the Atlantic Ocean off Angola and Namibiaas as shown by Rouault et al. (2003). Letelier et al. (2009) showed that in the Humboldt Current coastal region the cooling effect of upwelling is strongest in austral summer, which is consistent with the peak of upwelling-favourable wind in December and January. A poor simulation of the seasonal cloud and upwelling processes will contribute to the seasonality of SST biases in eastern boundary upwelling regions.

Most models have a seasonal warm SST bias in the eastern equatorial Atlantic (Figs. 1b-c, 2 and 3). The $T_{min}$ multi-model

mean bias can be more than 2°C larger than the $T_{max}$ multi-model mean bias. Richter and Tokinaga (2020) showed a similar seasonal warm bias in the CMIP6 multi-model mean, which is about 1-2°C larger during June-July-August than March-April-May. Richter et al. (2012) argued that the warm SST bias in the eastern equatorial Atlantic during June-July-August is linked to overly deep thermoclines caused by overly weak easterlies during March-April-May. Therefore, the warm bias can be attributed to overly weak easterlies in the CMIP6 multi-model mean (Li et al., 2020; Lyu et al., 2020). GISS-E2-1-G and GISS-E2-1-H

have the largest seasonality of SST warm bias in the eastern equatorial Atlantic, with $T_{min}$ biases up to 5°C. Richter and Tokinaga (2020) illustrated that warmer than observed SSTs in the equatorial Atlantic lead to excessive precipitation. Roxy (2014) quantified the SST-precipitation relationship: a 1°C SST increase corresponds to a 2 mm/day precipitation increase. Therefore, the 5°C $T_{min}$ warm bias in GISS-E2-1-G and GISS-E2-1-H could cause a 10 mm/day increase in precipitation.

Although the amplitudes of biases are different in $T_{max}$ and $T_{min}$, the global patterns and signs of $T_{max}$ and $T_{min}$ biases

are similar to each other in most models (Figs. 2, 3). Wang et al. (2014) indicated that the SST bias of the CMIP5 multi-model mean has a pattern independent of season but did not analyse the seasonality in bias in individual models. Our results show two exceptions: E3SM-1-0 and IPSL-CM6A-LR, which both have an overall warm bias in $T_{max}$, but an overall cold bias in $T_{min}$ (Figs. 2h,t 3h,t), which tend to cancel out in the annual means. The $T_{max}$ RMSE is 1.38°C for E3SM-1-0 and 1.36°C for IPSL-CM6A-LR, the $T_{min}$ RMSE is 1.39°C for E3SM-1-0 and 1.21°C for IPSL-CM6A-LR, whereas the $T_{mean}$ RMSE

is only 1.17 °C for E3SM-1-0 and 0.94°C for IPSL-CM6A-LR. In E3SM-1-0, the global annual average mixed layer depth is generally too shallow (Golaz et al., 2019), which can contribute to the summer SST warm bias and winter SST cold bias, and





a similar process may be affecting IPSL-CM6A-LR. These results illustrate the risks involved in assessing only annual means, as models may have greater biases than assumed, so tropical cyclone formation, for example, may be overpredicted.

In mid-latitudes the SST seasonal cycle is well represented by an annual sinusoid whereas in equatorial and polar regions an

annual sinusoid explains little of the total SST seasonal variance (Trenberth, 1983; Yashayaev and Zveryaev, 2001). In regions with fairly sinusoidal SST annual cycles such as the subtropics, models have realistic SST seasonal cycles with well simulated amplitude and phase of the annual cycle (Figs. 7c-d). Phase biases are mainly within 1 month (Figs. 4, 5, 6). In subtropical regions, seasonal SST biases are consistent with biases in $T_{mean}$. Differences between the $T_{max}$ and $T_{min}$ biases are smaller than those in non-sinusoidal regions (Fig. 7). In regions with non-sinusoidal SST seasonal cycles such as the western equatorial

Pacific, northwestern Indian Ocean, the Arctic and the Antarctic, models tend to have biases in amplitudes or phases of their SST seasonal cycles (Figs. 4, 5, 6, 7a-b,e-f).

In the western equatorial Pacific, the SST seasonal cycle in WOA18 is modest (less than 1°C), whereas in some models such as MPI-ESM1-2-HR, GISS-E2-1-G, GISS-E2-1-H and especially INM-CM5-0 the seasonal cycle is much larger (Fig. 7a). In INM-CM5-0, the $T_{cycle}$ is about 2°C and there is a cold SST bias throughout the year, reaching 3°C during September-

October-November (Fig. 7a). Similar to our analysis, Volodin et al. (2017) noted that INM-CM5-0 has a cold bias of more than 4°C in annual mean temperature in the upper 700 m of the western equatorial Pacific. The cold bias could limit the skills of models in simulation of El Niño/Southern Oscillation (ENSO) and ENSO-induced teleconnections. For example, a cold bias in the western equatorial Pacific results in a rising branch of the Walker circulation that is too far west in many coupled climate models leading to too weak ocean-atmosphere coupling and unrealistic ENSO dynamics (Bayr et al., 2018). The associated

convective response along the equator during ENSO events is too far west leading to westward shift in the sea level pressure response in the North Pacific and precipitation response in the subtropics (Bayr et al., 2019).

In the northwestern Indian Ocean where the monsoon system prevails, SST has a semi-annual cycle, but most models are unable to reproduce this with the correct amplitude and phase (Figs. 4, 5, 6, 7b). Most CMIP6 models have SST cold biases in this region throughout the year, while the biases are generally larger during March-April-May than other months and the multi-

model mean fails to simulate the primary maximum SST (Fig. 7b). Cold SST biases in the northwestern Indian Ocean lead to a significant reduction of the monsoon rainfall over the Indian subcontinent (Prodhomme et al., 2014; Levine and Turner, 2012). Thus the cold biases in the CMIP6 models are likely to lead to overly weak monsoon precipitation. Consistent with our result, McKenna et al. (2020) found a cold SST bias over the northwestern Indian Ocean in the CMIP6 multi-model mean. Fathrio et al. (2017) showed that the SST cold bias over the western Indian Ocean in the CMIP5 multi-model mean has a seasonal

cycle with the coldest SST bias occurring in April, whereas the coldest SST bias in our CMIP6 multi-model mean occurs in May. GISS-E2-1-G and GISS-E2-1-H fail to simulate a realistic second minimum SST in August (Fig. 7b), which would lead to overly intense tropical cyclones. SST in the northwestern Indian Ocean determines the onset of the summer monsoon (Sijikumar and Rajeev, 2012; Jiang and Li, 2011). The primary maximum SST is two months later in ACCESS-ESM1-5 than in WOA18 (Fig. 7b), which suggests a delayed summer monsoon onset in projections using that model.





## 3.2 Impact of model characteristics on SST seasonal extrema

We have shown that biases in $T_{max}$, $T_{min}$ and $T_{cycle}$ are different between models. We now use the diversity in the 20 CMIP6 models to explore the effects of different model characteristics on the magnitude of these biases as quantified by global area-weighted RMSE for $T_{max}$, $T_{min}$, $T_{cycle}$ and $T_{mean}$.

No significant correlation was found between the modelsâ™ seasonal biases and horizontal ocean resolution (supplementary Fig. S3). Chassignet et al. (2020) used four pairs of matched low-resolution and high-resolution ocean simulations from FSU-HYCOM, AWI-FESOM, NCAR-POP and IAP-LICOM to isolate the effect of ocean horizontal resolution, and compared their representation of global SST. They found that enhanced horizontal resolution does not deliver unambiguous SST bias improvement in all regions for all models, which is consistent with our finding. Nor did we find any correlation of seasonal biases with atmospheric resolution (supplementary Fig. S5), ocean grid type, ocean vertical coordinate, and inclusion (or not) of biogeochemical processes (circles or squares in Figs. 8 and 9).

The only characteristic yielding a statistically significant relationship was the ocean vertical resolution (Figs. 8, 9). The importance of vertical resolution for reducing seasonal biases is not unexpected: SST is influenced by ocean stratification and ocean vertical mixing processes, whose representation depends upon the vertical resolution. It has been found that high resolution in the upper ocean is important for the representation of diurnal and intraseasonal SST variability in ocean general circulation models (Misra et al., 2008; Xavier et al., 2008; Ge et al., 2017). Ideally we would have considered the number of vertical levels in the upper ocean. However, the number of vertical levels in the upper ocean (e.g. upper 200 m) cannot be unambiguously determined for models using an isopycnal or sigma vertical coordinate (6 out of 20 in our study) as their level depths vary with location and time (Bleck, 2002; Shchepetkin and McWilliams, 2005). Excluding the isopycnal and sigma models, the remaining high vertical resolution models are mainly from the Met Office Hadley Centre family, and hence any relationship between SST biases and vertical resolution in the upper ocean might have been overly influenced by that particular family. Hence we use the total number of vertical levels and top grid cell thickness (table 1) as proxies for the vertical resolution. Our study emphasises the importance of vertical resolution for simulating seasonal extreme SST and annual mean SST.

For the 20 models, there is a decrease in bias with increasing total number of vertical levels (Fig. 8). We calculated the inter-model correlation between global RMSE and total number of vertical levels following the method of Wang et al. (2014). The correlations are significant for $T_{max}$, $T_{min}$, and $T_{mean}$, with the largest correlation of -0.648 for $T_{max}$. The higher correlation between global $T_{max}$ RMSE and ocean vertical resolution is likely linked to shallower mixed layer depths in summer than in winter. RMSE is also correlated with top grid thickness (but with smaller correlation than total number of vertical levels): models with a smaller top grid thickness tend to have smaller biases (Fig. 9).

The impact of ocean vertical resolution on SST biases varies with latitude and season. Ocean vertical resolution is most important for $T_{max}$ at low latitudes (supplementary Figs. S1-2). SST biases decrease with number of vertical levels in the Benguela, Humboldt and California upwelling regions (supplementary Figs. S6-8). Only the Canary upwelling region, which has the smallest SST bias among the main four eastern boundary upwelling regions, does not have a good inter-model correlation between SST biases and ocean vertical resolution (supplementary Fig. S9).





**Figure 8.** Global RMSE of (a) $T_{max}$, (b) $T_{min}$, (c) $T_{cycle}$ and (d) $T_{mean}$, all against the total number of vertical levels in ocean. Circles represent earth system models, while squares represent non earth system models. The size of the markers represents the ocean horizontal resolution for that model, with larger markers for models with lower horizontal resolution. The black line is the line of best fit (with the least sum of squared errors). The inter-model correlation R is shown on each panel.





**Figure 9.** As in Fig. 8, but against the thickness of top grid in ocean.

## 4  Conclusions

Using the newly-released CMIP6 models, this study provides a global view of the biases in SST extrema, identifies regions with large seasonal bias, and suggests a future direction to reduce these biases. Global area-weighted $T_{max}$, $T_{min}$ and $T_{cycle}$ RMSEs are typically 1-2°C. Most models have $T_{max}$ and $T_{min}$ biases of the same sign at most locations, apart from IPSL-CM6A-LR and E3SM-1-0 which have an overall warm bias in $T_{max}$ and an overall cold bias in $T_{min}$. When averaged across the whole globe, the bias in $T_{mean}$ is typically consistent with $T_{max}$ and $T_{min}$ biases, but certain regions (eastern boundary

upwelling regions, polar regions, the eastern equatorial Atlantic, the North Pacific) show significant differences between winter and summer biases. Seasonal processes related to wind and cloud could be the main reasons for seasonal SST biases, but depend upon region. Further investigations of wind and cloud biases in CMIP6 models for different seasons could be undertaken to





better understand the causes of seasonal SST biases. In regions with non-sinusoidal SST seasonal cycles, models tend to have biases in amplitudes and/or phases of their SST seasonal cycles. For the models we examined, those with increased vertical

resolution in the ocean generally had a better representation of SST extrema, particularly $T_{max}$. This is likely related to the ability of the higher resolution models to better represent the surface mixed layer, and particularly shallow mixed layers in summer. For improving the accuracy of future climate projections, we suggest that as much priority (or possibly more) should be given to increasing vertical ocean model resolution as is given to increasing horizontal resolution.

*Code availability.*  All codes that support the finding of this study are available from YW, upon reasonable request.

*Author contributions.*  YW performed data analysis and prepared the manuscript under the supervision of KJH, DPS and GMD.

*Competing interests.*  The authors declare that they have no conflict of interest.

*Acknowledgements.*  We thank NOAA (National Oceanic and Atmospheric Administration), University of Hamburg and Met Office Hadley Centre for allowing access to the climatology data sets. We thank all modelling centres for carrying out CMIP6 simulations used here, and Earth System Grid Federation (ESGF) for archiving the data and providing access. This work was supported by the European Research

Council under the European Union's Horizon 2020 research and innovation programme (grant agreement n° 741120). YW was supported by the China Scholarship Council (grant agreement n° 201706310146). Computing and data storage resources were provided by JASMIN, the UK collaborative data analysis facility.



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
