# Peer review of "Seasonal extrema of sea surface temperature in CMIP6 models"

_Ocean Science, 2021_

## Author Response (AR1)

I find that the paper is a valid contribution and I find that the methodology applied is right. I have few minor comments that the authors will hopefully polish for next version of the paper.

We thank the reviewer for their helpful comments that help to improve the manuscript.

1. Page 2, line 58. The authors mention that they assign a 2°C threshold to the biases in Tmax and Tmin to exclude points from the analysis. I wonder whether they have previously performed some sensitivity analysis to analyze the impact of this threshold. I guess that such an error might be more or less important at points with higher or lower variances of SST. I think this should be checked.

Thank you for the comment. We had performed some sensitivity analysis before. The following figures are versions of Figure 1 showing biases of multi-model mean in (a) Tmean (b) Tmax (c) Tmin (d) Tcycle. Black dots mark grid points excluded from our analysis. Figures from left to right used 1°C, 2°C and 3°C thresholds to the maximum differences between the three climatologies, respectively.

[Figure]

We see that using 1°C threshold will exclude too large areas and many regions with large SST biases will be missing. The 3°C threshold mask covers a few coastal regions, however some points with high uncertainty (up to about 3°C) in the open ocean and coastal regions are not excluded from calculating the model bias. The mask using 2°C threshold mostly covers coastal areas, and it also includes a few regions in the Arctic, around ACC, Agulhas Current and Benguela Current. (Please notice that when plotting the figure, we interpolate the mask from 0.25°*0.25° grid into 2.5°*2.5° grid to make the black dots less dense.)

Threshold=1°C, percentage of the mask: 9% for Tmax, 6% for Tmin, 12% for Tcycle and Tmean

Threshold=2°C, percentage of the mask: 4% for Tmax, 3% for Tmin, 4% for Tcycle and Tmean

Threshold=3°C, percentage of the mask: 2% for Tmax, 1% for Tmin, 2% for Tcycle and Tmean

The seasonal cycle of monthly climatological SST from WOA18 is shown below, suggesting that most of the areas (except polar regions) covered by the masks with 2°C threshold have high variance of SST.

[Figure]

The maximum difference between any two of the three climatologies (WOA18, HadISST and WAGHC) for (a) Tmean (b) Tmax (c) Tmin and (d) Tcycle is shown in the figure below.

[Figure]

2. Page 4, line 70. "a small bias in a multi-model mean...individual models" seems like a little tautology to me.

Thank you for the comment. We will remove this sentence in the revised version.

3. I find that the RMSE values in figures 2, 3, 5 and 6 might be joined in a single table, so that a better analysis of systematic behaviors (if any) could be performed.

Thank you for the insightful suggestion. We plan to add a figure (Figure 7) comparing RMSE values in all models and the following text to the revised paper.

[Figure]

Figure 7. The global area-weighted RMSE of the biases in (a) Tmax, Tmin, Tmean and Tcycle (b) timing of Tmax and Tmin.

"In most of the models the global RMSE is larger in Tmax than in Tmin (Fig. 7a). As the bias in Tmax and Tmin is largely consistent with Tmean bias, Tcycle RMSE is small compared to Tmax and Tmin RMSEs in most models. Different biases in Tmax, Tmin, Tcycle and Tmean suggest that models have different performance in simulating SST seasonal variation and annual mean. The "best" and "worst" models depend on whether you choose SST seasonal extrema or annual mean as your metric. For example, GFDL-CM4 and HadGEM-GC31-MM have the smallest RMSE in Tmax and thus they are best for simulating tropical cyclones and heatwaves; SAM0-UNICON has the smallest RMSE in Tmin and thus it is best for simulating the properties of intermediate and deep waters."

"Models have different performance in simulating the timing of Tmax and the timing of Tmin. All the models except ACCESS-ESM1-5 have smaller global RMSE in the timing of Tmax than in the timing of Tmin (Fig. 7b). HadGEM3-GC31-MM has the smallest global RMSE in the timing of Tmax, whereas HadGEM3-GC31-LL and HadGEM3-GC31-MM have the smallest global RMSE in the timing of Tmin."

4. I find the palettes in Figures 5 and 6 difficult to grasp. Probably the fact that some isolated points are high in absolute value require a large scale. However, since most points are of a low absolute value, it is difficult to grasp the differences from one map to another. I suggest that perhaps the authors should use a non-equally spaced palette to increase resolution in lower values.

Thank you for the helpful suggestion. We have updated the color bars in Figures 5 and 6 to emphasize the low values on the maps. The following figure shows one of the panels in Figure 5 with the old color bar (left) and new color bar (right).

[Figure]

5. In Figure 7, I suggest some labels referring to regions are added to the individual panels to improve readability "WestEqPac" in panel "a)", "NWIndOc" in "b)" and so on.

Thank you for the suggestion. We have added labels on the panels in Figure 7. An example panel is shown below.

[Figure]

6. Page 12, line 120. I am not sure that the increase in storminess can be assigned to heat fluxes into the storms but separated from increased atmospheric baroclinicity (Kushnir, 2002), not explicitly mentioned by the authors. I think this point must be revised.

Thank you for the comment. Our original statement seems wrong for the extratropics. Brayshaw et al. (2008) and Nakamura et al. (2004) show that a change in SST gradient in certain key regions impacts the storm track. That is, an SST bias may impact the SST gradient and thus impact storm tracks. Given the uncertainty that the change of SST gradient could be in any direction for any given bias depending upon the background state, here we remove "Models with a warm bias in Tmin are likely to generate overly intense winter storms, as warm SSTs will increase the storm energy source. Greeves et al. (2007) demonstrated that there was a clear link in the Hadley Centre models between winter SST warm bias to the east of Japan and increased storm intensity in the region."

Brayshaw, David James, Brian Hoskins, and Michael Blackburn. "The storm-track response to idealized SST perturbations in an aquaplanet GCM." *Journal of the Atmospheric Sciences* 65.9 (2008): 2842-2860.

Nakamura, Hisashi, et al. "Observed associations among storm tracks, jet streams and midlatitude oceanic fronts." *Earth's Climate: The Ocean–Atmosphere Interaction, Geophys. Monogr* 147 (2004): 329-345.

7. Page 12, lines 126-129. I guess that Myers et al. (2021) is a good reference to support the authors' hypothesis here.

Thank you for your recommendation. We have cited Myers et al. (2021) as "The large cold biases at northern hemisphere high latitudes in BCC-CSM2-MR, BCC-ESM1, GISS-E2-1-G and GISS-E2-1-H, are typically 2-5◦C smaller in Tmin than in Tmax (Figs. 2, 3, 7g-h). These cold biases are likely to be linked to cloud biases due to the cooling radiative effect of low cloud (Myers et al., 2021). The negative cloud radiative forcing is excessive in BCC-CSM2-MR (Wu et al., 2019) and BCC-ESM1 (cloud simulation likely to be similar to BCC-CSM2-MR), while overestimated low-cloud cover in GISS-E2-1-G and GISS-E2-1-H (Kelley et al., 2020) blocks more of the incoming solar radiation."

8. Page 14, lines 179-181. I suggest the authors to fit a simple sinusoidal signal here (period T=12 months) and the fraction of variance explained would allow the authors to show which areas respond to one or the other case.

Thank you for the suggestion. We have fitted sinusoids to the time series in Figure 7 (see below) and the following discussion has been added in the text.

"In regions with fairly sinusoidal SST annual cycles such as the subtropics (sinusoidal signal explains 87% of the observed variances in subtropical Northern Hemisphere and 89% of the observed variances in subtropical Southern Hemisphere), models have realistic SST seasonal cycles with well simulated amplitude and phase of the annual cycle (Fig. 7c-d)."

"In regions with non-sinusoidal SST seasonal cycles such as the western equatorial Pacific, northwestern Indian Ocean, the Arctic and the Antarctic (sinusoidal signal explains 33%, 23%, 58% and 46% of the observed variances), models tend to have biases in amplitudes or phases of their SST seasonal cycles (Figs. 4,5,6,7a-b,e-f)."

[Figure]

Kushnir, Y., Robinson, W. A., Bladé, I., Hall, N. M. J., Peng, S., & Sutton, R. (2002). Atmospheric GCM Response to Extratropical SST Anomalies: Synthesis and Evaluation, Journal of Climate, 15(16), 2233-2256. https://journals.ametsoc.org/view/journals/clim/1 5/16/1520-0442_2002_015_2233_agrtes_2.0.co_2.xml

Myers, T.A., Scott, R.C., Zelinka, M.D. et al. Observational constraints on low cloud feedback reduce uncertainty of climate sensitivity. Nat. Clim. Chang. 11, 501–507 (2021). https://doi.org/10.1038/s41558-021-01039-0

Review of "Seasonal extrema of sea surface temperature in CMIP6 models" by Wang et al.

This is an account of seasonal sea surface temperature (SST) biases in CMIP6 models with respect to the observed World Ocean Atlas climatology for the period 1981-2010. The authors document and discuss discrepancies between the multi-model mean as well as individual model seasonal cycles, and observations. They find model diversity in cold and warm season SST biases, as well as annual mean biases and seasonal cycle amplitude biases. Model bias is shown to increase with decreasing vertical ocean model resolution, highlighting promising future model development activities. These findings have important implications for the model specific assessment of regional climate projections. As such, the present study will a valuable reference in the climate model evaluation literature.

I find the manuscript to be interesting, relevant, and well-written. The presented results are novel and important within the ocean modeling community. The paper is therefore a good fit for Ocean Science. After some mostly minor concerns of mine have been addressed, I think this manuscript will make a welcome addition to the scientific literature.

We thank the reviewer for the helpful and constructive comments that helped us in improving our manuscript.

Main concern:

A somewhat substantial concern of mine is the use of the first realisation of each model for drawing fundamental conclusions about the model's biases. Individual ensemble members are subject to substantial chaotic climate variability, which might as well influence the seasonal cycle of SST. To address this issue, I suggest the authors compare r1 to the ensemble mean of individual models (or r2) to get an idea about how strongly SST seasonal cycles depend on the realisation of the models. This would make their analysis more rigorous and give their reported results more weight.

Thank you for your suggestion. We expect differences within each model ensemble to be small as we are examining thirty-year means. We have compared r1 with r2 for the SST seasonal cycles. The differences of Tmax, Tmin, month of Tmax and month of Tmin between r1i1p1f1 and r2i1p1f1 in each model are shown below (Figs 1-4). r1i1p1f1 and r2i1p1f1 are compared when available; r1i1p1f3 and r2i1p1f3 are compared for HadGEM3-GC31-MM and HadGEM3-GC31-LL; r1i1p1f2 and r2i1p1f2 are compared for UKESM1-0-LL. There are no results for SAM0-UNICON and GFDL-CM4 as they have only one ensemble member.

The differences of Tmax and Tmin between two ensemble members are within 0.5$^\circ$C over most of the global ocean, which are very small compared with the model biases of Tmax and Tmin (Figs. 1-2) for the ensemble member considered in our paper. As for the months of Tmax and Tmin, there are no differences between the two ensemble members over most of the global ocean, while one month differences exist in some specific regions (Figs. 3-4). The small differences between ensemble members demonstrate that SST seasonal cycles do not depend on the realisation of the models, and thus the model biases we report are robust.

The following sentence has been added to line 74 in the manuscript.

"To test the dependence of the biases found on the realisation of models, we compared the first and second ensemble members (except for SAM0-UNICON and GFDL-CM4 as they have only one ensemble member). The differences between ensemble members are very small compared with the model biases (supplementary Figs. S1-4), and thus the model biases we report are robust."

[Figure]

Figure 1. (Left) Differences between two ensemble members for Tmax. (Right) Tmax model biases for the ensemble member considered in our paper (Figure 2 in the paper, reproduced here for ease of comparison). Black dots mark grid points excluded from our analysis.

[Figure]

Figure 2. As in Fig.1, but for Tmin.

[Figure]

Figure 3. As in Fig. 1, but for the month of Tmax.

[Figure]

Figure 4. As in Fig. 1, but for the month of Tmin.

Specific comments:

l. 3 Should these "commonly used climatologies" be defined here?

Thank you for the comment. We have rewritten "commonly used climatologies" as "commonly used climatologies (WOA18, WAGHC and HadISST)".

l. 9 I suspect that the authors report "no significant relationship" of SST extrema bias "with horizontal ocean model resolution", but suggest the authors be specific about that.

Thank you for the suggestion. We have rewritten the sentence as "No significant relationship of SST seasonal extrema with horizontal ocean model resolution is found."

l. 21 Suggestions: move the abbreviation "CMIP5" into the brackets and the long form out of the brackets.

Thank you for the suggestion. We have updated the manuscript accordingly.

ll. 31-32 Drawing conclusions about projected seasonal cycles from the analysis presented in this paper assumes i) stationarity of the biases, and ii) that the biases are consistent between r1 (analysed here) and the ensemble mean (used for projections) of the individual models. While i) is difficult to test and could (should?) be discussed as a caveat, I think that ii) requires some attention in the revision of the manuscript (see my main concern above).

Thank you for the helpful suggestions.

For i), we have added the following sentences on lines 269.

"If there is a substantial change in the climate, it should be considered that the pattern of biases in Tmax and Tmin may change."

For ii), we have compared r1 with r2 for the SST seasonal cycles (details are in the response to the main concern above), and the following sentence has been added to line 74 in the manuscript.

"To test the dependence of the biases found on the realisation of models, we compared the first and second ensemble members (except for SAM0-UNICON and GFDL-CM4 as they have only one ensemble member). The differences between ensemble members are very small compared with the model biases (supplementary Figs. S10-14), and thus the model biases we report are robust."

ll. 32-34 I do not see the value of a table of contents here and thus suggest deleting it.

Thank you for the comment. We have removed the sentences.

ll. 41-42 The extra information about INM-CM5-0 (Volodin, pers. comm.) could be placed more appropriately in Table 1. Please consider moving it accordingly.

Thank you for the comment. We have moved the sentences to the footnote of Table 1.

ll. 51-52 I wonder if the averaging of T values from different months in case of a shifted seasonal cycle would be problematic. Could the authors please comment?

Thank you for your suggestion. To calculate multi-model mean, we averaged SST seasonal extrema (Tmax and Tmin) in 20 models and indeed in some regions seasonal extrema occur in different months in different models. However, we don't see it as a serious problem as the timing of seasonal extrema is very similar in most of the global ocean in most models, which is suggested by the small bias (within one month in most of the global ocean in most models) in the timing of seasonal extrema (see Figs 5-6 in the manuscript). Although in some cases we calculated multi-model mean by averaging SST in different months, we cannot pick some specific months for averaging to avoid this problem as SST seasonal extrema occur in different months in different models. In fact, picking specific months would also assume stationarity and it is probably worse. If Tmax and Tmin were to get earlier or later in future climate projections, people could get erroneous results if they look at particular months.

l. 53 To avoid potential confusion, I suggest being specific here that the RMSE of the four different T quantities is calculated against observations for global SST.

Thank you for your suggestion. We have rewritten the sentence "we calculated the area-weighted root mean square error of the model against WOA18 (henceforth RMSE) for global SST." as "we calculated the area-weighted root mean square error of Tmax, Tmin, Tmean and Tcycle of the model against WOA18 (henceforth RMSE) for global SST."

ll. 60-61 Where are the excluded grid points typically located? Maybe add a sentence about that here.

Thank you for the suggestion. The following sentence has been added on line 59.

"The excluded grid points are mostly located in coastal areas, a few regions in the Arctic, and around the ACC, Agulhas Current and Benguela Current."

l. 77 I cannot make out to what the "it" at the start of this sentence refers. Please be specific.

Thank you for the comment. We have rewritten the sentence "It demonstrates that seasonal cycles in CMIP6 models are out of phase with observations." as "The bias in the timing of Tmax and Tmin demonstrates that the seasonal cycles in CMIP6 models are out of phase with observations.".

l. 80 Explicitly stating once in this paragraph that high latitudes show a larger bias than low latitudes would make the entire story easier to follow. Please consider this addition.

Thank you for the suggestion. We have added the following sentence on line 94.

"High latitudes show larger biases than low latitudes."

l. 97 I think it should be "...salinity biases in the Arctic."

Thank you for pointing this out. We have corrected it in the revised version.

Figure 7 The different x axes of a-f compared to h-i are somewhat confusing to me. The message of this figure might be easier to grasp if the figure was split in two separate figures.

Thank you for the suggestion. We have split Figure 7 in two separate figures as you suggested.

ll. 104-105 The two occurrences of "is larger" in this sentence lack clarity without a reference (larger than what?). I recommend using absolute language such as "shows a maximum".

Thank you for the comment. We have rewritten the sentence as "The latent heat loss shows a maximum in summer (Yu, 2007), while the ocean heat advection shows a maximum in winter when meridional SST gradients are greatest."

ll. 131-132 Which part of the cloud is underestimated? From the sentence itself, it is unclear if it is cloud cover, formation, ...

Thank you for the comment. We have rewritten the sentence as "In most models there is a warm Tmean bias in the Southern Ocean, commonly attributed to excessive short wave radiation linked to cloud process representation deficiencies."

l. 150 Separate "Namibia" and "as"

Thank you for pointing this out. We have corrected it in the revised version.

ll. 229-230 What is the significance level at which this correlation is significant? Which significance test was used? Without this information, the statement of significance is not worth much. Similarly, I suggest adding p-values to figures 8 and 9.

Thank you for the suggestion. We have added p-values to figures 8 and 9. When $p < 0.05$, we can say that the relationship between global RMSE and ocean vertical resolution is significant. The sentence "The correlations are significant for Tmax, Tmin, and Tmean, with the largest correlation of -0.648 for Tmax." has been rewritten as "The relationship between SST biases and total number of vertical levels is significant for Tmax, Tmin and Tmean (p-values$<0.05$), with the largest correlation of -0.648 for Tmax."